# CO_2_/CH_4_ Separation in Amino Acid Ionic Liquids, Polymerized Ionic Liquids, and Mixed Matrix Membranes

**DOI:** 10.3390/molecules29061357

**Published:** 2024-03-19

**Authors:** Gowri Selvaraj, Cecilia Devi Wilfred

**Affiliations:** 1Centre of Research in Ionic Liquids (CORIL), Institute of Contaminant Management (ICM), Universiti Teknologi PETRONAS, Seri Iskandar 32610, Perak, Malaysia; selgowri@gmail.com; 2Fundamental and Applied Sciences, Universiti Teknologi PETRONAS, Seri Iskandar 32610, Perak, Malaysia

**Keywords:** amino acid ionic liquids, polymerized ionic liquids, mixed matrix membranes, CO_2_ separation

## Abstract

The ability to efficiently separate CO_2_ from other light gases using membrane technology has received a great deal of attention due to its importance in applications such as improving the efficiency of natural gas and reducing greenhouse gas emissions. A wide range of materials has been employed for the fabrication of membranes. This paper highlights the work carried out to develop novel advanced membranes with improved separation performance. We integrated a polymerizable and amino acid ionic liquid (AAIL) with zeolite to fabricate mixed matrix membranes (MMMs). The MMMs were prepared with (vinylbenzyl)trimethylammonium chloride [VBTMA][Cl] and (vinylbenzyl)trimethylammonium glycine [VBTMA][Gly] as the polymeric support with 5 wt% zeolite particles, and varying concentrations of 1-butyl-3-methylimidazolium glycine, [BMIM][Gly] (5–20 wt%) blended together. The membranes were fabricated through photopolymerization. The extent of polymerization was confirmed using FTIR. FESEM confirmed the membranes formed are dense in structure. The thermal properties of the membranes were measured using TGA and DSC. CO_2_ and CH_4_ permeation was studied at room temperature and with a feed side pressure of 2 bar. [VBTMA][Gly]-based membranes recorded higher CO_2_ permeability and CO_2_/CH_4_ selectivity compared to [VBTMA][Cl]-based membranes due to the facilitated transport of CO_2_. The best performing membrane Gly-Gly-20 recorded permeance of 4.17 GPU and ideal selectivity of 5.49.

## 1. Introduction

The global demand for the use of natural gas as a cleaner and more efficient fuel is persistently increasing. The worldwide consumption of natural gas is estimated to reach 182 trillion cubic feet by 2030. Natural gas primarily consists of CH_4_. It also contains substantial quantities of impurities including water, CO_2_, H_2_S, N_2_, and other hydrocarbons. The removal of acid gases such as CO_2_ and H_2_S is one of the crucial steps in natural gas treatment before it is transported to pipelines or stored into portable cylinders as compressed natural gas. The presence of more than 2.5% CO_2_ will affect the ability of the natural gas to burn efficiently and safely. Separating CO_2_ from natural gas (1) increases the fuel heating value, (2) decreases the volume of gas to be transported in pipelines and cylinders, (3) reduces pipeline corrosion, and (4) prevents atmospheric pollution [1]. Therefore, the development of efficient methods for capturing CO_2_ is considered to be of major importance [2]. A regulation for CO_2_ emissions entails the development of specific CO_2_ capture technologies that can be retrofitted to existing systems as well as designed into new plants to achieve 90% of CO_2_ capture, while limiting the increase in the cost of electricity to no more than 35% [3]. Currently utilized natural gas purification techniques involve absorption of CO_2_ using solvents like amine and hot aqueous potassium carbonate solution, pressure swing adsorption, and membrane technology. Despite the full maturity of the gas absorption processes using alkanolamine solvents, they suffer from solvent degradation, corrosion problems, and high parasitic energy consumption during solvent regeneration [4]. The performance of gas separation materials, particularly those based on graphene, metal-organic frameworks (MOFs), and polymers, has shown significant promise. Graphene oxide (GO)-composite membranes have been widely investigated for better selectivity and permeability of CO_2_ due to the presence of a strong conjugated π system within these membranes [5]. MOFs have also been extensively studied for CO_2_ capture, CH_4_ recovery, and olefin/paraffin separation. MOF membranes demonstrate superior molecular sieving capabilities based on differences in chemical affinity and pore size [6]. Polymers, on the other hand, remain the primary choice for commercial membranes due to their ease of processing and low costs. Their limitations in terms of gas permeability and selectivity can be overcome by integrating MOF or graphene components within the polymer matrix to create mixed matrix membranes.

The application of polymeric membranes for CO_2_ separation from natural gas received a breakthrough in the 1980s. Cellulose acetate (CO_2_/CH_4_ selectivity of 19) membranes are still widely used [7,8,9]. They have been applied to natural gas sweetening where the concentrations of CO_2_ and H_2_S contained in high-pressure natural gas should be lowered to the levels of meeting the gas pipeline specifications (CO_2_ < 2% and H_2_S < 4 ppm) [1]. Cellulose acetate membranes have been receiving stiff competition from newer membranes, such as polyimides and perfluoropolymers. Despite the numerous advantages of current commercial membranes, these systems are blamed to perform at lower efficiency than amine systems for acid gas removal for several reasons, such as the presence of contaminants, the concentration polarization permeability/selectivity trade-off, physical aging, and plasticization [10].

Innovative membranes capable of high CO_2_ permeance and selectivity at the appropriate operating conditions are promising alternatives to meet these challenges. In the past few years, various membranes have been developed which target the attractive area above the so-called “Robeson upper bound” [11]. Hybrid membranes combining ionic liquids (ILs) and a polymeric support is a rather new concept developed in the early 2000s. ILs have a unique combination of physicochemical properties such as high CO_2_ solubility (mole basis), high selectivity, low volatility, and designable structure to adjust chemical/physical properties [11]. Using 1-ethyl-3-methyl imidazole as the cation and 20 natural amino acids as the anions, Ohno et al. synthesized AAILs [12]. At equilibrium, the CO_2_ absorption capacity of tetrabutylphosphonium amino acid [P4444]^+^[AA]^−^ is 50 mol% of ILs [13]. The amino acid anion has shown more excellent CO_2_ capture in ILs than the [BF_4_]^−^, Cl^−^, [NO_3_]^−^ and [PF_6_]^−^ anions of the same cation. Brennecke et al. also chose a series of AAILs to absorb CO_2_.They found that the amine groups in amino acid anions were better at capturing CO_2_ than those in amine cation-based ILs [14]. Polymerized ionic liquids (PILs) are a family of functional polymeric materials with ionic liquid groups covalently attached to a polymer backbone, i.e., pendant PILs, or covalently connected to form a functional polymer chain, i.e., backbone PILs [15]. Numerous options of ionic liquid units along with different polymer architectures enable PILs with different physical and chemical properties to satisfy multiple applications. PILs are commonly ammonium-based polymerized ionic liquids, namely poly(p-vinylbenzyltrimethyl ammonium tetrafluoroborate), P[VBTMA][BF_4_], poly([2-(methacryloyloxy)ethyl] trimethyl ammonium tetrafluoroborate), P[MATMA][BF_4_] [16], and a range of amine-derived ionic liquids bearing vinyl and ally groups [17]. Shahrom et al. reported that the polymerization of ionic liquids derived from vinylbenzenyltrimethylammonium (VBTMA) and L-glycine could lead to a rise in uptake of CO_2_. The amino acid ionic liquids (AAILs) typically contain amine groups which are capable of forming hydrogen bonds with CO_2_. The affinity of the amino acids will absorb the CO_2_ molecule and form complexes with AAILs through hydrogen bonds [18]. Hence, we are proposing a series of composite membranes comprising PILs together with amino acid-based ILs. Due to their polymer macrostructure, PIL membranes have improved processability, durability, and mechanical stability. The non-polymerized ILs suspended in the PIL matrix have exhibited enhanced gas separation. The non-polymerized IL serves as a plasticizer that increases the diffusivity of CO_2_. A much stronger Coulombic attraction is anticipated between the polymeric support and the IL [19]. Bara et al. reported 20 mol% of the non-polymerized IL within the polymer matrix increased CO_2_ separation by 100–250% relative to the neat polymeric membrane [20]. An increase in IL percentage can lead to higher permeability without increasing or decreasing selectivity.

Recently, another promising class of membrane material for CO_2_/CH_4_ separations are mixed matrix membranes (MMMs) that consist of a porous solid (such as a zeolite) in a polymer matrix. MMMs were proposed as a strategy to utilize the excellent separation properties of zeolites into a more easily processable material [21,22]. Zeolites are a class of porous, crystalline silicate-based frameworks with pore sizes matched to the size of light-gas molecules [23,24]. In this study, zeolite particles, PILs consisting of [VBTMA][Gly] or [VBTMA][Cl], and non-polymerized ILs consisting of [BMIM][Gly], were combined to produce MMMs. [BMIM][Gly] is a functionalization IL with amino acids that enables facilitated transport, thus offering both enhanced CO_2_ permeability and selectivity. A photopolymerization technique was used to prepare these composite membranes in the presence of divinylbenzene and 2-hydroxy-2-methylpropiophenone. The concentration of [BMIM][Gly] varied from 5 to 20 wt%. FTIR was used to obtain structural information of the PILs and [BMIM][Gly] composite membranes. CO_2_/CH_4_ permselectivity studies were performed at room temperature and with a feed side pressure of 2 bar.

## 2. Results

### 2.1. Characterization of [VBTMA][Gly] and [BMIM][Gly] Ionic Liquids

The anion exchange and neutralization steps yielded the desired amino acid ionic liquids. The ^1^H-NMR spectra for [VBTMA][Gly] and [BMIM][Gly] are provided in Appendix A. The ^1^H-NMR analysis confirmed the structures of the ILs synthesized. **(Vinylbenzyl)trimethylammonium glycine [VBTMA][Gly]** δ 3.14 (s, 9H), 3.16 (s, 2H), 4.56 (s, 2H), 5.39 (d, 1H, 8.8 Hz), 5.94 (d, 1H, 14 Hz), 6.79–6.84 (m, 1H), 7.56 (d, 2H, 6.4 Hz), 7.63 (d, 2H, 6.4 Hz); **1-Butyl-3-methylimidazolium glycine [BMIM][Gly]** δ 0.87 (t, 3H, 7.3 Hz), 1.21 (m, 2H), 1.75 (m, 2H), 2.73 (s, 2H), 3.88 (s, 3H), 4.20 (t, 2H, 7.2 Hz), 7.80–7.87 (m, 2H), 9.53 (s, 1H). The ^1^H NMR of [VBTMA][Gly] had a chemical shift at 3.16 ppm indicating the presence of CH_2_ from the glycinate anion [25]. This showed that the anion exchange and neutralization with the glycine amino acid to glycinate ionic liquids was successful. For [BMIM][Gly], the CH_2_ peak from glycinate appeared at 2.73 ppm. The VBTMA appears to have more deshielding effect than imidazolium towards glycinate, causing its CH_2_ peak to appear more downfield.

Figure 1 shows the FTIR spectra of the ILs involved. The region at 3300 cm^−1^ showed the presence of N–H amides. The region at 3000–2800 cm^−1^ represents C–H stretching. These peaks are present in all three spectra. The peak produced at 1560 cm^−1^ is a result of asymmetric COO^−^ stretching. The intensity of the band is due to the strong dipole moment of the carbon-oxygen bonds [18]. This peak only arose in the spectra for [VBTMA][Gly] and [BMIM][Gly] due to the presence of the amino acid. A weak peak at 1475 cm^−1^ is due to C=C in the aromatic ring. The distinct peak at 1381 cm^−1^ is from the C–N group. The vinyl CH=CH_2_ peaks of [VBTMA] Cl and [VBTMA][Gly] show absorption between 975 to 885 cm^−1^.

### 2.2. Characterization of Mixed Matrix Membranes

#### 2.2.1. FTIR Analyses

The composition of the mixed matrix membrane comprising [VBTMA][Cl] or [VBTMA][Gly] with a varied concentration of [BMIM][Gly] and 5 wt% zeolite is summarized in Table 1.

For [VBTMA][Cl] as the polymerized ionic liquid, Cl-Gly-0 is the abbreviation where Cl denotes 2.00 g [VBTMA][Cl] as the polymerized ionic liquid, Gly denotes [BMIM][Gly], and 0 denotes the mass of [BMIM][Gly]. The composition of the mixed membrane includes 5 wt% zeolite. For [VBTMA][Gly] as the polymerized ionic liquid, Gly-Gly-0 is the abbreviation where the former Gly denotes 2.00 g [VBTMA][Gly] as the polymerized ionic liquid, the latter Gly denotes [BMIM][Gly], and 0 denotes the mass of [BMIM][Gly]. The composition of the mixed membrane includes 5 wt% zeolite.

Spectra of the mixed matrix membrane comprising [VBTMA][Cl] or [VBTMA][Gly] with a varied concentration of [BMIM][Gly] with 5 wt% zeolite (WZ) are shown in Figure 2 and Figure 3. All the FTIR spectra showed C=C bending between 975 and 885 cm^−1^. This is an indication that complete polymerization has not taken place. Bara et. al. reported that with 30 min of irradiation, >90% conversion of vinyl groups can be achieved [20]. However, irradiating the sample longer resulted in membranes that are more fragile and less flexible. For [BMIM][Gly], a strong peak arose at 1577 cm^−1^ corresponding to asymmetric CO_2_ stretching of carboxylic acid. This peak is absent in the pure [VBTMA][Cl]-based membrane. The intensity of these peaks increases with the concentration of [BMIM][Gly]. Cl-Gly 5 with 5 wt% loading of [BMIM][Gly] showed a very weak presence of this peak due to the insufficient amount of [BMIM][Gly] for the equipment to detect. All composites containing Gly-Gly shown in Figure 3 give rise to the COO^−^ peak. The intensity of the peaks is higher due to the presence of the dual glycine-based ILs. The higher amino acid content is anticipated to contribute to higher CO_2_ permeance through facilitated transport.

Figure 4 shows FTIR spectra of zeolite particles, Cl^_^Gly 10, and Gly^_^Gly 10. Both the membranes contain the same amount of zeolite particles. The broadband at 3450 cm^−1^ and a peak at 1637 cm^−1^ correspond to the OH stretching and bending modes of water molecules absorbed by the zeolite, respectively [26,27]. Besides, the band that appears at 973 cm^−1^ is attributed to the Si–O–Al asymmetric stretching vibration of T–O bonds, where T refers to tetrahedrally bonded Si or Al. Other significant bands at 547 and 660 cm^−1^ are respectively ascribed to the double six-membered rings of T–O–T symmetric stretching and Si–O–Si symmetric stretching [28]. Also, the band at 460 cm^−1^ refers to the symmetric bending of T–O modes. Bands arising from water absorbed by zeolite overlap with N–H amides and the COO^−^ band stemming from the ILs, as seen in the Cl-Gly-Zeolite and Gly-Gly-Zeolite spectra. Both membranes showed the presence of a 974 cm^−1^ band. Membranes with zeolite also gave rise to very weak peaks at 553 and 709 cm^−1^.

#### 2.2.2. Morphology Studies of MMMs

Figure 5 and Figure 6 display the cross-sectional images of the [VBTMA][Cl]- and [VBTMA][Gly]-based membranes at 300× magnification. The MMMs produced are dense in structure without pores. The interfacial adhesion seems to be good enough to allow proper embedding of the particles into the polymeric phase, which results in the absence of large interfacial voids. The zeolite particles are randomly distributed within the membrane matrix. In these samples, a greater number of isolated particles and small agglomerates were observed. This behaviour could be explained by the strong hydrogen bonding between the zeolite particles and the polymer matrix [29]. The hydroxyl groups of the zeolite particles strongly interact with both the hydroxyl and carbonyl groups of the amino acid in the IL [30]. Increasing the content of [BMIM][Gly] did not significantly affect the morphology of these membranes.

#### 2.2.3. Thermal Analysis of MMMs

Figure 7 shows the TGA curves of [VBTMA][Cl]- and [BMIM][Gly]-based membranes. All thermograms indicate the presence of moisture in the membranes. About 10% weight loss occurs around 100 °C due to the evaporation of water. Amino acid-based ILs are hygroscopic; hence, they tend to attract and hold on to water molecules. The PIL and the non-polymerized IL decompose around 300 °C. The calculated average weight loss is 58%. Decomposition of the ILs appears to transpire in a single step based on the thermograms. According to Mathew et al., decomposition of P[VBTMA][Cl] starts at 230 °C and ends at 370 °C [31]. [BMIM][Gly], on the other hand, begins to decompose around 210.35 °C [32]. For example, P[VBTMA][Cl] in the Cl-Gly-0 membrane decomposed at 284 °C. The addition of 5 wt% [BMIM][Gly] increased the decomposition temperature by 30 °C. However, further increments of [BMIM][Gly] to 15 and 20 wt% resulted in a reduced decomposition temperature of 303 and 289 °C, respectively. Reduced intermolecular forces between the IL and the PIL and enhanced segmental motion of the polymer matrix could be the reason behind the reduced thermal stability with increasing non-polymerized IL content [33]. Another significant weight loss which accounts for about 31% weight loss takes place at 415 °C. This weight loss can be credited to the decomposition of zeolite particles. According to Rykl and Pechar, the crystal lattice of zeolite changes at 260 °C and it decomposes into an amorphous phase above 470 °C [34]. The zeolite particles in Cl-Gly-0 decomposed at 410 °C. The decomposition temperature increases to 415 and 419 °C with the incorporation of 5 and 10 wt% of [BMIM][Gly], respectively. Contradictorily, the membranes with 15 and 20 wt% of the non-polymerized IL recorded slightly lower decomposition temperatures for zeolite at 417 and 418 °C, respectively. The T_g_ data is summarized in Table 2.

Figure 8 shows the TGA curves of [VBTMA][Gly]- and [BMIM][Gly]-based membranes. [VBTMA][Gly]-based membranes have shown to contain more moisture than [VBTMA][Cl]-based membranes. The TGA data indicates about 15% weight loss around 100 °C. This could be due to an increased amount of glycine in the membrane. Weight loss associated with decomposition of the polymeric support and the non-polymerized IL also occurs at a lower temperature than [VBTMA][Cl]-based membranes. This accounts for about 60% weight loss. For example, the PIL and IL in the Gly-Gly-0 membrane decompose at 192 °C as shown in Figure 8. Based on our previous results, P[VBTMA][Gly] decomposes at 192 °C [35]. Increasing the concentration of [BMIM][Gly] resulted in a slight decrease in the thermal degradation temperature. Membranes with 5, 10, 15, and 20 wt% of [BMIM][Gly] experienced this weight loss at 186, 183, 182, and 181 °C, respectively. The decomposition temperature of zeolite particles in [VBTMA][Gly]-based membranes is higher than that in [VBTMA][Cl]-based membranes. The zeolite particles in Gly-Gly-0 decomposed at 439 °C. The inclusion of 5 wt% of [BMIM][Gly] resulted in a reduction in thermal stability (436 °C). Increasing the content of the IL to 10 and 15 wt%, on the other hand, increased the decomposition temperature to 437 and 440 °C, respectively. The degradation temperature once again reduced to 437 °C at 20 wt% of the non-polymerized IL. In general, the higher thermal stability could be an indication of higher interfacial adhesion between the zeolite particles and the glycine-based ILs. The thermal degradation temperature is summarized in Table 3.

The glass transition state (T_g_) refers to the temperature at which a material changes state from a rigid glass-like solid to a flexible rubbery compound [36]. The T_g_ of pure P[VBTMA][Cl] is 150 °C [37], for P[VBTMA][Gly] it is −46.50 °C [35], and for [BMIM][Gly] it is −41.15 °C [38]. All fabricated membranes recorded T_g_ values in the temperature range of 87 to 90 °C, as tabulated in Table 1. In the case of [VBTMA][Cl]-based membranes, the addition of [BMIM][Gly] and zeolite particles resulted in a decrease in the T_g_ value from the base polymer. The addition of a very low-T_g_ additive diluted the polymer matrix and thus decreased the T_g_ [39,40]. A decreased T_g_ correlates to improved polymer chain flexibility. The incorporation of 5 to 15 wt% of [BMIM][Gly] further reduced the T_g_. This can be attributed to the plasticization of the polymer chains in the presence of the non-polymerized IL [41]. Nonetheless, increasing the content of [BMIM][Gly] to 20 wt% increased the T_g_. At this concentration, the IL limited the flexibility of the polymeric chain. Both Cl-Gly-0 and Gly-Gly-0 membranes recorded very similar T_g_ values. However, [VBTMA][Gly]-based membranes exhibited an increased T_g_ with the addition of the non-polymerized IL and zeolite particles, compared to the base polymer. The same anion on the PIL and IL could mean increased interaction between the IL and the polymeric matrix, thus reducing the mobility of the polymer chain. Besides this, glycine is a bulkier anion than chloride. Consequently, the [VBTMA][Gly]-based polymeric support is more rigid. The addition of [BMIM][Gly] reduced the T_g_ of the [VBTMA][Gly]-based membranes. Yet, a clear trend cannot be established between the T_g_ and increasing the concentration of the non-polymerized IL. Li et al. reported that the T_g_ of MMMs increases about 1.5 to 3 °C due to the incorporation of zeolite particles [42]. Zeolite tends to restrict the movement of the polymer chains by the formation of hydrogen bonding between the zeolite particles and the polymer [43]. The thermal degradation temperature is summarized in Table 4.

### 2.3. CO_2_ Permeability and CO_2_/CH_4_ Selectivity Studies

The measured permeabilities of CO_2_ and CH_4_ in [VBTMA][Cl]- and [VBTMA][Gly]- based MMMs at room temperature and with a feed side pressure of 2 bars are shown in Figure 9 and Figure 10, respectively. [VBTMA][Gly]-based membranes performed better than [VBTMA][Cl]-based membranes in terms of CO_2_ permeability. Even in the case of samples without [BMIM][Gly], the [VBTMA][Gly]-based membrane recorded 39% higher permeance than its counterpart with [VBTMA][Cl]. Based on our results, it is clear that increasing the concentration of [BMIM][Gly] resulted in enhanced CO_2_ permeation. For example, the highest CO_2_ permeation was recorded by Gly-Gly-20 at 4.17 GPU.

The facilitation of CO_2_ transport is accomplished by the “carrier” inside a facilitated transport membrane, which can reversibly react with CO_2_ [44]. The amino acid serves as the “carrier” in this case. At the feed side interface of the membrane, CO_2_ reacts with glycine and forms a CO_2_-glycine complex, which diffuses along its concentration gradient to the permeate side of the membrane. Due to a lower CO_2_ partial pressure on the permeate side, CO_2_ is released from the complex to the permeate side, while regenerating the amino acid that can then react with another CO_2_ molecule on the feed side.

In a previous study, we fabricated the polymerized ionic liquids with non-polymerized ionic liquids without zeolite and obtained lower CO_2_ permeability. Both groups of membranes in this study were shown to have a higher affinity towards CO_2_, proving that the addition of the zeolite powder increases the permeability of carbon dioxide. Consistently lower CH_4_ permeation was recorded. Zeolite plays an important role in reducing CH_4_ permeation. The reason is that its molecular structure with a pore size around 0.38 nm is suitable for CO_2_/CH_4_ gas separation application [45,46]. Both Cl-Gly-0 and Gly-Gly-0 allowed for more CH_4_ to permeate than any of the membranes containing [BMIM][Gly]. For instance, Gly-Gly-0 and Gly-Gly-5 recorded a CH_4_ permeation of 1.93 and 0.66 GPU, respectively. Hence, it is evident that the added [BMIM][Gly] also plays a role in improving the selectivity of these membranes towards CO_2_. CO_2_ has a strong quadrupole moment compared to the non-polar CH_4_. Therefore, CO_2_ has a stronger interaction with the amino acid-based IL than CH_4_, further improving selectivity.

Gly-Gly-20 recorded the highest selectivity, followed by Gly-Gly-5, as displayed in Figure 11. However, it is suggested that having too much of [BMIM][Gly] too close to the pores of the non-polymerized ionic liquids can jeopardize the diffusion of the CO_2_, as the pores could be blocked. CH_4_ permeation did not follow a clear trend like CO_2_ concerning [BMIM][Gly] concentration, which resulted in the ideal selectivity not being linear. While it has no attraction towards ionic liquids, it is still able to diffuse through the zeolite pores better, which hold better to the polymerized ionic liquids, with increasing [BMIM][Gly]. Cl-Gly-20 had a 60% lower selectivity set side by side with Gly-Gly-20. This proves the point that having two AAILs improves CO_2_ permeability and CO_2_/CH_4_ selectivity.

## 3. Materials and Methods

### 3.1. Materials

(Vinylbenzyl)trimethylammonium chloride [VBTMA][Cl] 99.0%, 1-butyl-3-methylimidazolium chloride [BMIM][Cl] 99.0%, glycine, and the anion exchange resin (Amberlyst A-26 Hydroxide, OH^−^ form) purchased from Sigma Aldrich (Kuala Lumpur, Malaysia), Malaysia were used to synthesize the ionic liquids. Methanol and acetonitrile used for synthesis and purification were purchased from Merck (Kuala Lumpur, Malaysia). Divinylbenzene 85%, 2-hydroxy-2-methylpropiophenone 97%, and zeolite were also purchased from Sigma Aldrich, Malaysia to prepare the photopolymerized membranes. All chemicals were used without any further purification.

### 3.2. Synthesis of [VBTMA][Gly] and [BMIM][Gly] Ionic Liquids

#### 3.2.1. Anion Exchange from Cl^−^ to OH^−^

A chromatography column was packed with 48 g of the anion exchange resin Amberlyst A-26 (OH^−^ form). 10 g (0.0472 moles) of [VBTMA][Cl] was dissolved in 200 mL of methanol (or ethanol/water) until it completely dissolved. The [VBTMA][Cl] solution was passed through the resin. The resulting [VBTMA][OH] was collected dropwise at a flow rate of one drop every 10–15 s. The presence of residual Cl^−^ ions were tested using acidified silver nitrate solution (AgNO_3_). The formation of brown precipitate instead of white precipitate confirmed the absence of Cl^-^ ions. A similar procedure was used to convert [BMIM][Cl] to [BMIM][OH]. The [VTMA][OH] and [BMIM][OH] were light yellow liquids.

#### 3.2.2. Neutralization Using Amino Acids

An equimolar amount of glycine was added to the [VBTMA][OH] solution to neutralize it. The mixture was allowed to stir at room temperature for 24 h. The resulting IL was dried using rotary evaporation. The excess amino acid was precipitated using a 3:7 ratio of methanol and acetonitrile. The resultant white precipitate was removed through gravity filtration. The solvent mixture used for purification was separated from the IL through rotary evaporation. The washing process was repeated as required. The resulting (vinylbenzyl)trimethylammonium glycine [VBTMA][Gly] was vacuum dried overnight and stored in a desiccator. The synthesis of [VBTMA][Gly] is summarized in Figure 12. The above-mentioned procedure was repeated using equimolar [BMIM][OH] and glycine to synthesize 1-butyl-3-methylimidazolium glycine, [BMIM][Gly]. The [VTMA][Gly] was obtained in a 72% yield as a yellow solid. [BMIM][Gly] was obtained in an 87% yield as a yellow solid.

#### 3.2.3. Characterization of [VBTMA][Gly] and [BMIM][Gly]

^1^H NMR was measured with a Bruker 400MHz spectrometer using MeOD as a solvent. FTIR spectra were recorded using Perkin Elmer Spectrum One. The measurements were made between the ranges of 4000 to 400 cm^−1^.

### 3.3. Fabrication of Mixed Matrix Membranes through Photopolymerization

An amount of 2 g (0.0080 moles) of [VBTMA][Gly], [BMIM][Gly], and 5 wt% of zeolite was weighed out into an 8 mL vial. An amount of 1 g of methanol was added to the mixture. The mixture was stirred gently for a day. After that, the sample was degassed in an ultrasonic bath for 1 h. Amounts of 5 mol% of divinylbenzene (cross-linker) and 1 wt% of 2-hydroxy-2-methylpropiophenone (photoinitiator) were added to the IL solution. The mixture was mixed using a mechanical shaker for 10 min. The casting solution was poured on a piece of quartz plate treated with Rain-X. Spacers with a thickness of 50 μm were added to the edges of the plate and the second piece of quartz plate was slowly placed on top of the casting solution. The sample was irradiated using a 6 W 365 nm UV lamp for 1 h. This procedure was repeated to fabricate [VBTMA][Cl]-based membranes. The concentration of [BMIM][Gly] in the composite membranes varied, as shown in Table 1.

### 3.4. Characterization of Mixed Matrix Membranes

FTIR spectra were recorded using Perkin Elmer Spectrum One. The measurements were made between the ranges of 4000 to 400 cm^−1^. The morphology of the membranes was studied using Variable Pressure Field Emission Scanning Electron Microscope (VPFESEM) model Zeiss Supra55VP. The surface and cross-sectional morphologies of the membranes were determined to determine whether the membranes are porous or nonporous. The magnification used was 1.00 Kx to 10.00 Kx, the working distance was 3–6 mm, and the accelerating voltage was 2 to 5 kV. Thermal properties and degradation temperature (Td) were measured using the thermogravimetric analyzer (TGA) model Perkin Elmer Pyris 1, with a heating rate of 10 °C/min and a flow rate of nitrogen at 20 mL/min. The temperature range used in this study was 50 to 800 °C. The glass transition temperature (T_g_) was measured by differential scanning calorimetry (DSC), using a Mettler Toledo model DSC 1 with a heating rate of 10 °C/min and a flowrate of nitrogen at 20 mL/min in the temperature range from −90 to 250 °C. It was run in three cycles with heating to 250 °C, cooling to −90 °C, and heating back to 250 °C.

### 3.5. CO_2_/CH_4_ Permeability and Selectivity Studies

CO_2_/CH_4_ separation performances of these membranes were studied using a single gas permeation unit at room temperature and with a feed side pressure of 2 bar. The schematic diagram of the permeation unit is shown in Figure 13.

The membrane samples were cut into circles with a diameter of 4.7 cm and placed in the membrane cell. Inside the membrane cell, the membranes were supported with a perforated polypropylene support and a porous nylon membrane. Then, an O-ring was placed on top of the membrane before the unit was sealed to prevent gas leakage from the module. The feed side pressure was regulated using a mass flow controller. The flow rate was kept constant at 100 mL/min for all the experiments. The downstream side pressure was assumed to be at atmospheric pressure. Before commencing the permeation test, the system was vacuumed to remove trapped gases and impurities in the unit. The permeate flow rate was recorded every 15 min for 3 times using a bubble flow meter. The amount of gas permeating the membrane was calculated using Equation (1).
(1)$$\frac{P_{i}}{l}=\frac{Q_{i}}{T_{i}.A.∆p_{i}}$$
where P_i_ is the permeance of gas i, Q_i_ is the volumetric flow rate of the permeate gas (cm^3^/s), T_i_ is the absolute temperature (K) (298.15 K), A is the effective area of the membrane (cm^2^) (A = πr^2^), ∆p_i_ is the differential partial pressure across the membrane (cmHg), and l is the thickness of the membrane (cm). Permeance is expressed in the gas permeation unit, which is defined by 1GPU = 1 × 10^−6^ × (cm^3^(STP)/cm^2^∙cmHg∙s). The ideal selectivity of the membranes was calculated using Equation (2).
(2)$$\alpha =\frac{P_{{CO}_{2}}}{P_{{CH}_{4}}}$$
where α is ideal selectivity, $P_{{CO}_{2}}$ is the permeability of CO_2_, and $P_{{CH}_{4}}$ is the permeability of CH_4_.

## 4. Conclusions

The effect of amino acid functionalization on both the polymeric support and the non-polymerized IL, as well as on CO_2_ permeability and CO_2_/CH_4_ selectivity, was successfully studied. This was carried out by developing three-component MMMs with [VBTMA][Cl] or [VBTMA][Gly], [BMIM][Gly], and zeolite particles. The structure of the MMMs was confirmed using FTIR. The morphology of the membranes was analyzed using FESEM. The membranes produced are dense in structure without voids and pores. TGA showed that all membranes contained a significant amount of moisture due to the presence of the amino acids. The PIL and IL decompose before the zeolite particles. All membranes have a T_g_ between 87 and 90 °C.

Based on CO_2_ permeability and CO_2_/CH_4_ selectivity studies, Gly-Gly-20 performed the best, with a CO_2_ permeability of 4.17 GPU and an ideal selectivity of 5.49. Compared to Cl-Gly-20, this is a 50% improvement in terms of CO_2_ permeability and a 59% increment in terms of ideal selectivity. Increasing the content of [BMIM][Gly] also contributes to a higher CO_2_ permeability. The CO_2_ permeability of [VBTMA][Gly] shows a clear increase in permeability with the increasing concentration of [BMIM][Gly].

## Figures and Tables

**Figure 1 molecules-29-01357-f001:**
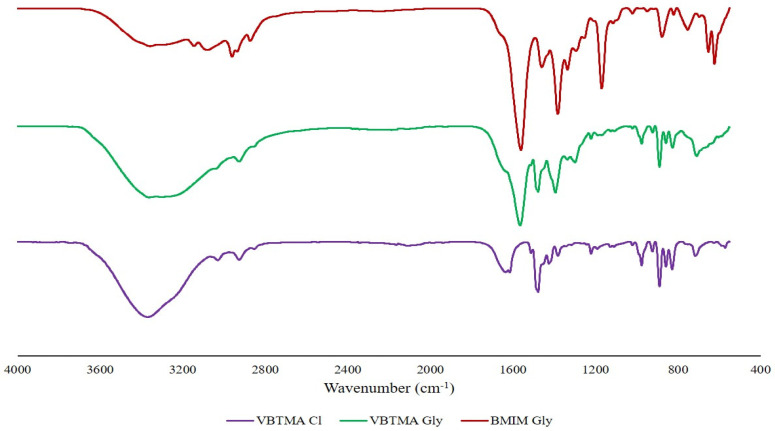
FTIR spectra of [VBTMA][Cl], [VBTMA][Gly], and [BMIM][Gly].

**Figure 2 molecules-29-01357-f002:**
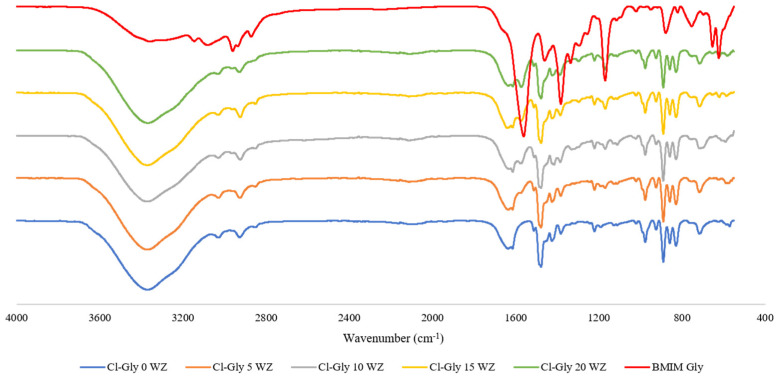
FTIR spectra of [VBTMA][Cl]^_^ and [BMIM][Gly]^_^based membranes.

**Figure 3 molecules-29-01357-f003:**
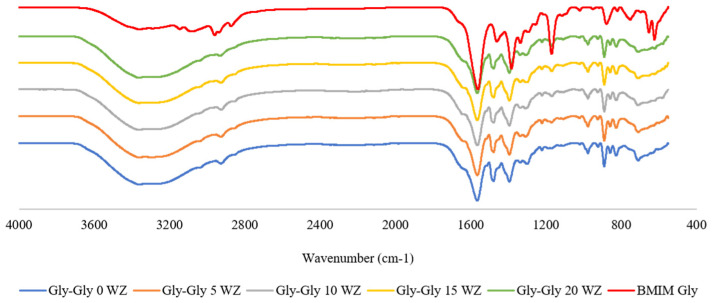
FTIR spectra of [VBTMA][Gly]^_^ and [BMIM][Gly]^_^based membranes.

**Figure 4 molecules-29-01357-f004:**
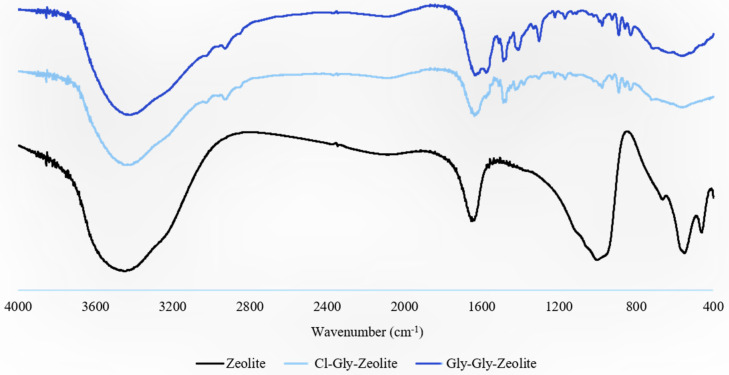
FTIR spectra of zeolite, Cl^_^Gly^_^Zeolite, and Gly^_^Gly^_^Zeolite.

**Figure 5 molecules-29-01357-f005:**
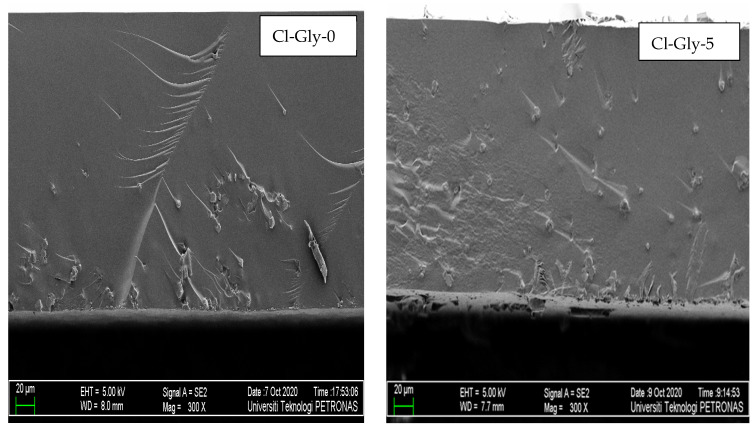
FESEM images of [VBTMA][Cl]- and [BMIM][Gly]-based membranes.

**Figure 6 molecules-29-01357-f006:**
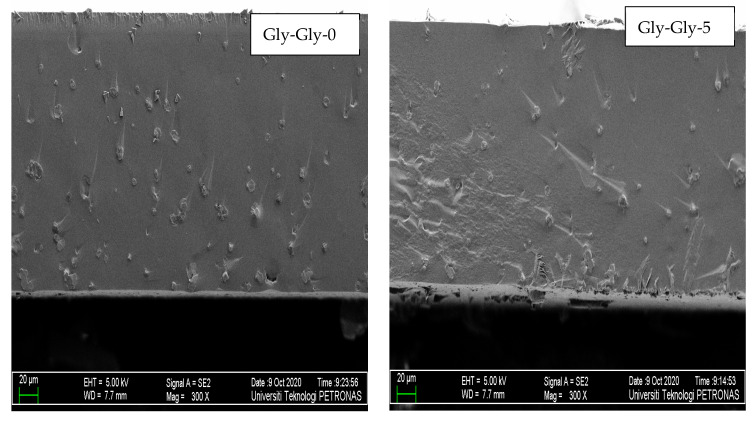
FESEM images of [VBTMA][Gly]- and [BMIM][Gly]-based membranes.

**Figure 7 molecules-29-01357-f007:**
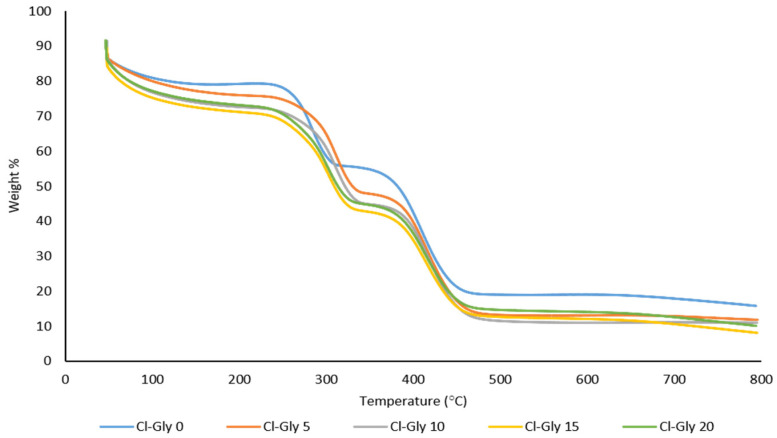
TGA curves of [VBTMA][Cl]- and [BMIM][Gly]-based MMMs.

**Figure 8 molecules-29-01357-f008:**
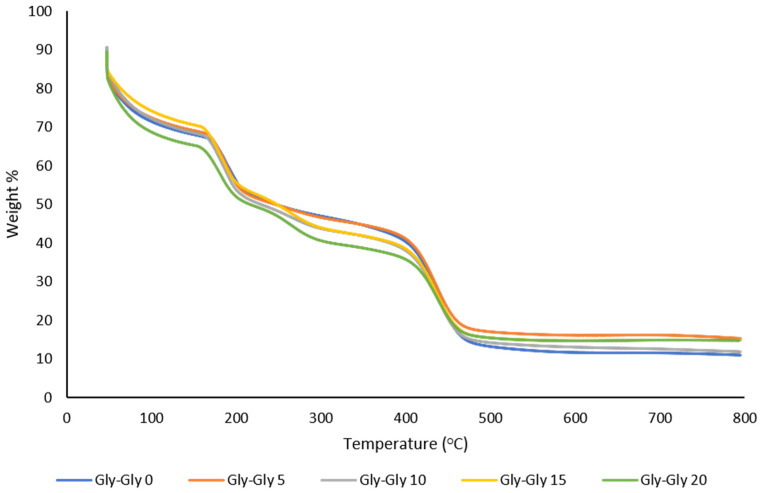
TGA curves of [VBTMA][Gly]- and [BMIM][Gly]-based MMMs.

**Figure 9 molecules-29-01357-f009:**
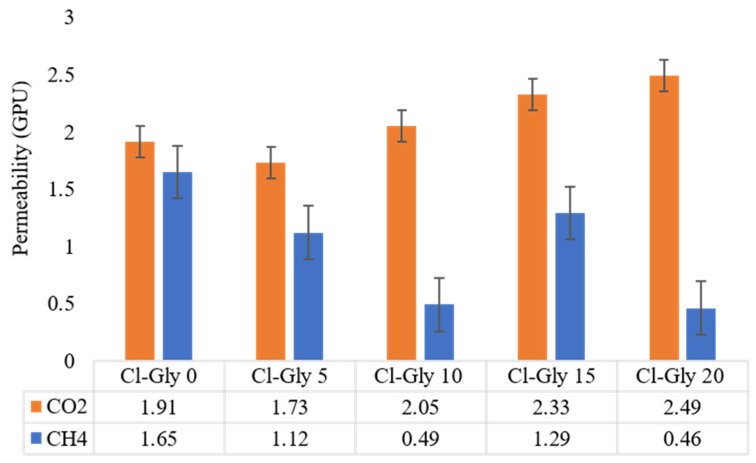
CO_2_ and CH_4_ permeability through [VBTMA][Cl]-based membranes.

**Figure 10 molecules-29-01357-f010:**
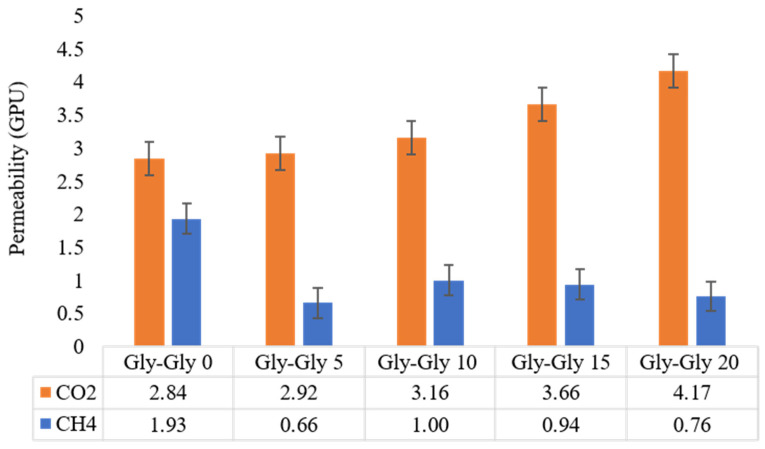
CO_2_ and CH_4_ permeability through [VBTMA][Gly]-based membranes.

**Figure 11 molecules-29-01357-f011:**
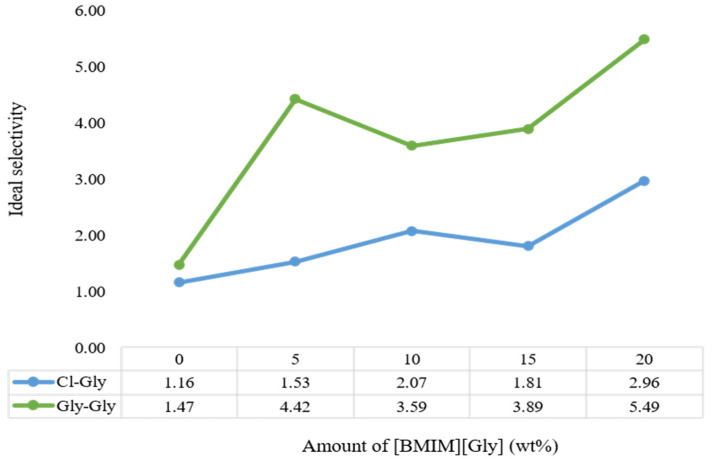
Ideal selectivity of [VBTMA][Cl]- and [VBTMA][Gly]-based MMMs at varying concentration of [BMIM][Gly].

**Figure 12 molecules-29-01357-f012:**
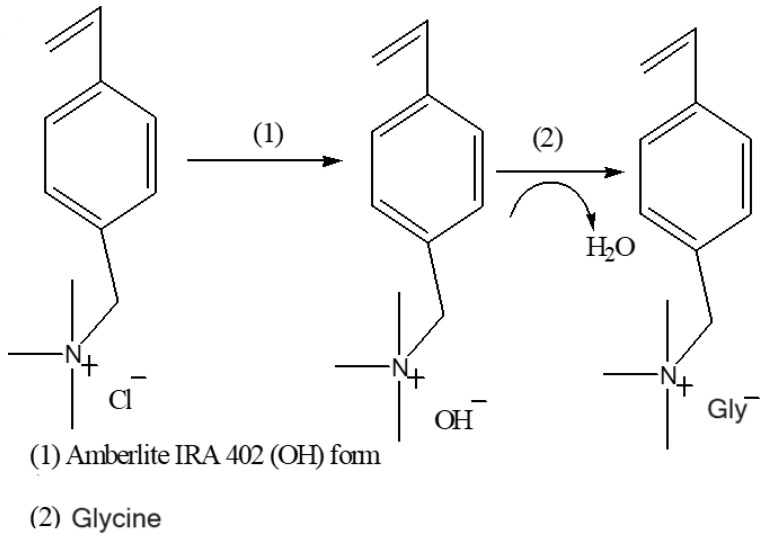
Synthetic route for [VBTMA][Gly].

**Figure 13 molecules-29-01357-f013:**
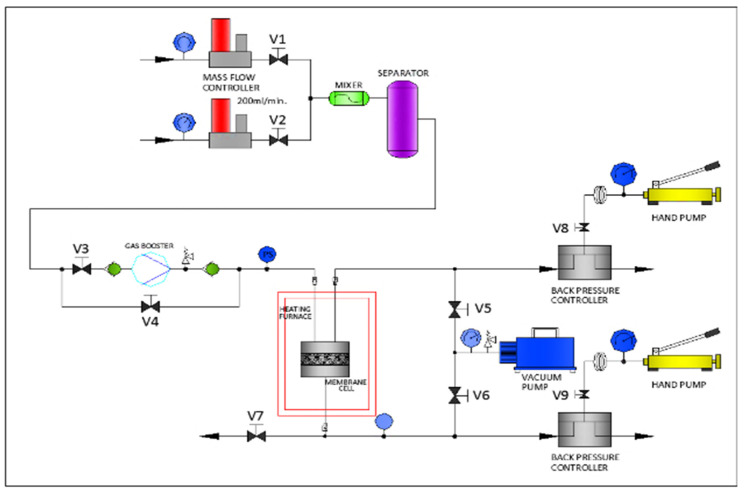
Schematic diagram of the permeation unit.

**Table 1 molecules-29-01357-t001:** Composition of fabricated mixed matrix membranes.

Membranes	[VBTMA][Cl] (g)	[VBTMA][Gly] (g)	[BMIM][Gly] (g)	Zeolite (g)
Cl-Gly-0	2.00	0.00	0.00	0.10
Cl-Gly-5	2.00	0.00	0.10	0.10
Cl-Gly-10	2.00	0.00	0.20	0.10
Cl-Gly-15	2.00	0.00	0.30	0.10
Cl-Gly-20	2.00	0.00	0.40	0.10
Gly-Gly-0	0.00	2.00	0.00	0.10
Gly-Gly-5	0.00	2.00	0.10	0.10
Gly-Gly-10	0.00	2.00	0.20	0.10
Gly-Gly-15	0.00	2.00	0.30	0.10
Gly-Gly-20	0.00	2.00	0.40	0.10

**Table 2 molecules-29-01357-t002:** Thermal degradation of poly[VBTMA]Cl ionic liquids, [BMIM][Gly], and mixed matrix membranes.

	Degradation Temperature, T_g_ (°C)	Degradation Temperature, T_g_ (°C)
Poly[VBTMA]Cl	230	
[BMIM]Gly	210.35 °C	
Membranes		
Cl-Gly-0	284	410
Cl-Gly-5	314	415
Cl-Gly-10	316	419
Cl-Gly-15	303	417
Cl-Gly-20	289	418

**Table 3 molecules-29-01357-t003:** Thermal degradation of the poly[VBTMA]Gly ionic liquids, [BMIM][Gly], and mixed matrix membranes.

	Degradation Temperature, T_g_ (°C)	Degradation Temperature, T_g_ (°C)
Poly[VBTMA]Gly	192	
[BMIM]Gly	210.35	
Membranes		
Gly-Gly-0	192	439
Gly-Gly-5	186	436
Gly-Gly-10	183	437
Gly-Gly-15	182	440
Gly-Gly-20	181C	437

**Table 4 molecules-29-01357-t004:** Glass transition state (T_g_) of [VBTMA][Cl]- and [VBTMA][Gly]-based membranes.

Membrane	T_g_ (°C)
Cl-Gly-0	89.90
Cl-Gly-5	89.82
Cl-Gly-10	89.28
Cl-Gly-15	89.19
Cl-Gly-20	90.23
Gly-Gly-0	89.44
Gly-Gly-5	88.41
Gly-Gly-10	89.15
Gly-Gly-15	89.34
Gly-Gly-20	87.57

## Data Availability

Data are contained within the article and Appendix A.

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
