# Peer review of "CO_2_/CH_4_ Separation in Amino Acid Ionic Liquids, Polymerized Ionic Liquids, and Mixed Matrix Membranes"

_molecules, 2024, doi:10.3390/molecules29061357_

Round 1
Reviewer 1 Report
Comments and Suggestions for Authors
The manuscript reports the synthesis and characterization of mixed-matrix membranes for CO2/CH4 separation. The membranes consisted of a porous solid (zeolite) with the polymerized ionic liquids [VBTMA][Gly] or [VBTMA][Cl], along with free IL [BMIM][Gly]. Every step of the experimental work, including the synthesis of the ionic liquids, fabrication of the mixed-matrix membranes with different amounts of [BMIM][Gly], and evaluation of CO2 permeability and CO2/CH4 selectivity, was carefully conducted. The results indicate that the mixed-matrix membrane [VBTMA][Gly] with 20% wt of [BMIM][Gly] exhibited the best performance in terms of CO2 permeability and CO2/CH4 selectivity. The discussion and conclusions drawn were supported by the experiments.
This manuscript reports on significant insights for the community. Consequently, I recommend publishing this manuscript in its current form.
Just some minor typos:
1) Page 2, lines 51-52
I didn't understand the structure of the sentence “Cellulose acetate (CO2/CH4 selectivity of 19 membranes are still widely used [5-7].”
2) Page 2, Line 73
“Hence”, instead of “_Hence”
3) Page 13, Table 2
From the reading of the manuscript, I have the feeling that the names of columns [VBTMA][Gly] and [VBTMA][Cl] are swapped.
4) Page 15, line 420
“CO\_2”, instead of “CO2” (It is missing the subscript in the second CO2)
Reviewer 2 Report
Comments and Suggestions for Authors
This paper discussed the fabrication of mix-matric membrane with zeolites, ionic liquid, and polymer, showing good selectivity towards CO2/CH4 gases. I suggest that this paper could be published on Molecule with further modifications:
1. The author should compare the performance with gas separation materials, such as graphene based membranes, MOFs, polymers, etc., and their drawbacks and benefits towards.
2. The author should add control experiments to compare the performance to polymer matrix itself, and polymer with zeolites to verify.
3. Gly-Gly 20 seems to give the best performance. How will the performance be influenced if further increase the concentration of [BMIM][Gly]?
4. CO2 permeability is improved with the increasing of [BMIM][Gly]. However, there is no obvious trend for CH4. Please explain more on this phenomenon.
Reviewer 3 Report
Comments and Suggestions for Authors
After carefully reviewing the manuscript, I believe that a major revision is necessary to address several significant issues and improve the overall quality of the work. The manuscript shows promise, but there are notable areas that require attention and refinement before it can be considered for publication. I have provided detailed feedback and suggestions for improvement in the attached review.
1. Please update the literature review about the ionic liquids with different polymerizable groups, such as Journal of Molecular Liquids, 283, 2019, Pages 427-439, (doi.org/10.1016/j.molliq.2019.03.061); New J. Chem., 2020,44, 12274-12288 (doi.org/10.1039/D0NJ00303D).
2. The introduction is well written and discusses important information that is needed for the justification of the work undertaken. However, there is lack of information about amino acids, not only as part of ionic liquids, but also about their ability to absorb CO2. This should be addressed also.
3. I do not think that the reference “free ILs” should be used. I understand that the authors would like to highlight that the additional ionic liquid is not polymerized, so maybe a better phrase would be just “non-polymerized IL”. I will not push on my proposition, but I would think about other options.
4. Line 131 is incoherent. “All composites containing Gly-Gly shown in Figure 3 showed , give rise to the COO- peak.”
5. Characterization of [VBTMA][Gly] and [BMIM][Gly]. It's crucial to consider J couplings in NMR analysis as they provide valuable information about molecular structure and connectivity. Including coupling constants ensures a comprehensive understanding of the compound's chemical environment, aiding in accurate structural confirmation and analysis.
6. Synthesis of [VBTMA][Gly] and [BMIM][Gly] ionic liquids.
a) In synthetic practice, it's common to provide not only the mass but also the moles and equivalents of reagents, often denoted within brackets. This comprehensive approach ensures clarity and precision in the preparation of chemical reactions.
b) The synthesis lacks details regarding color, yield and the physical form of the resulting compounds.
c) If equimolar amounts were used, where did the excess amino acid come from?
d) Figure 12. If the authors only obtain Glycine-based ionic liquids, why in the final product there is Arg- present?
e) Figure 12. Structures of chemical compounds need to be “cleaned up”.
7. Figure 4. Lines of different shades of gray are making it difficult to distinguish between them.
8. Table 2, composition of the membranes. I think it would benefit the manuscript cohesion if the information about the composition would be placed in the results and discussion section, rather than in the experimental section down below. Also, I do not fully understand the labels of the membranes. For example, the Cl-Gly compositions. Form the information in the table 2 I gather that the only components in this matrix are [VBTMA][Gly] and the varying concentrations of [BMIM][Gly]. I think the labels are wrong, and should be reversed.
9. Table 2. I would think that below the table footnote should be added, explaining the composition labels.
10. TGA results would be clearer and more readable if they were in a table in addition to be discussed in the text. Also, would consider adding to the table results from TGA for just ionic liquids and their mixtures, it would allow for easier comparison of the results.
11. Have the authors conducted any long-term stability studies to assess the durability and performance sustainability of the membranes?
12. How do the results of this study compare to existing literature on mixed matrix membranes for gas separation applications?
13. Have the authors investigated the influence of operating conditions, such as temperature and pressure, on the separation performance of the membranes?
14. Lines 368-370 it says “…(Tg) was measured by differential scanning coulometric (DSC)…”. It is Differential Scanning Calorimetry, not Coulometric. Also was the DSC analysis conducted only for 1 cycle or more?
Comments on the Quality of English Language
The manuscript should undergo thorough grammar and spell checking to ensure clarity and professionalism.
Round 2
Reviewer 2 Report
Comments and Suggestions for Authors
The paper is appropriate for publication after adding control experiments.
Reviewer 3 Report
Comments and Suggestions for Authors
After the appropriate revisions made by the authors I recommend this manuscript for publication.